# Multi-modal Few-shot learning: A benchmark

**Frederik Pahde**[1,2]**, Moin Nabi**[1]**, Tassilo Klein**[1]
[1] Machine Learning Research, SAP SE, Berlin, Germany
[2] Humboldt Universität Berlin
{frederik.pahde, m.nabi, tassilo.klein}@sap.com

## Abstract

The state-of-the-art deep learning algorithms generally require large amounts of data for model training. Lack thereof can severely deteriorate the performance. To this end, we propose a multi-modal approach that facilitates bridging the information gap by means of meaningful joint embeddings. Specifically, we present a benchmark that is multi-modal during training (i.e. images and texts) and single-modal in testing time (i.e. images), with the associated task to utilize multi-modal data in base classes (with many samples), to learn explicit visual classifiers for novel classes (with few samples). Next, we propose an framework built upon the idea of cross-modal data hallucination. In this regard, we introduce a discriminative text-conditional GAN for sample generation with a simple self-paced strategy for sample selection. Experiments on our proposed benchmark demonstrate that learning generative models in a cross-modal fashion facilitates few-shot learning by compensating the lack of data in novel categories.

## 1 Introduction

In recent years, deep learning techniques have achieved exceptional results in many domains such as computer vision and NLP. These advances can be explained by improvements to algorithms and model architecture along with increasing computational power, and in particular growing availability of big data. However, the big data assumption, which is key for deep learning applications, is at the same time the limiting factor. For many applications, it is often too expensive or even impossible to acquire enough training samples in order to learn a model at sufficient accuracy. Furthermore, the requirement for large amounts of training data is in stark contrast to human learning, which can quickly learn from few instances. This makes alternative learning approaches that require less training data an attractive research topic. Thus research in the domain of few-shot learning, i.e. learning and generalizing from few training samples, has gained more and more interest (e.g. Ravi & Larochelle (2017), Snell et al. (2017), Vinyals et al. (2016)). However, research conducted has mainly focused on approaches with data coming from only one modality, especially images. Overcoming this limitation and including data from other modalities, e.g. textual descriptions in addition, can further improve the model. The assumption of our approach is that having fine-grained descriptions provided from multi-modal data can force the model to focus on the more discriminative features (e.g., parts and attributes) of novel classes in order to achieve improved performance in the few-shot learning setting (Elhoseiny et al. (2017)). This assumption leads to the proposed study of few shot learning with multi-modal data, more precisely images with fine-grained textual descriptions. The principal contribution of this paper is to extend few-shot learning to deal with multimodal data. We address this problem from a cross-modal generative perspective, combining ideas from meta-learning which have been put forward in Hariharan & Girshick (2017).

The most closely related work to the proposed approach is by Hariharan & Girshick (2017) and Wang et al. (2018), who similarly use hallucinated data for few-shot learning with the difference of the restriction to a single-modal image context. Analogously, Zhang et al. (2017) and Reed et al. (2016b) proposed to use Generative Adverserial Network (Goodfellow et al. (2014)) to generate images from textual descriptions. They, however, just employed it in a zero-shot fashion, ignoring few number of samples available of novel classes. Our contribution includes: **First**, a benchmark for multimodal few-shot learning based on the challenging fine-grained visual recognition task. **Second**, a class-discriminative text-conditional generative adversarial network (tcGAN) that facilitates

the few-shot learning by hallucinating images conditioned on fine-grained textual descriptions, featuring robustness and outperforming the baseline in the challenging low-shot scenario.

## 2 METHOD

**A Multimodal Few-shot Learning Benchmark:** The goal is to build a benchmark for multimodal few-shot learning that mimics situations that arise in practice. To this end, we proposed a few-shot learning benchmark inspired by Hariharan & Girshick (2017) and extend it to work with multimodal data. We split the classes $C$ into base classes $C_{Base}$ for which many samples exist and novel classes $C_{Novel}$ with just a few samples. In this scenario, data from $C_{Base}$ is used to learn meaningful representations in order to perform few-shot learning on $C_{Novel}$. Our proposed benchmark is multimodal in training and single-modal in testing. To that end, the task is to utilize multimodal data of training set, to learn explicit visual classifiers for novel classes. The training samples are tuples $x_j = (I_j, T_j)$ consisting of an image $I_j \in \mathcal{I}$ and a textual description $T_j \in \mathcal{T}$, where $\mathcal{I}$ and $\mathcal{T}$ denote the image space and text space, respectively. In test time, however, the model is tested only with the image data from $C_{Novel}$.

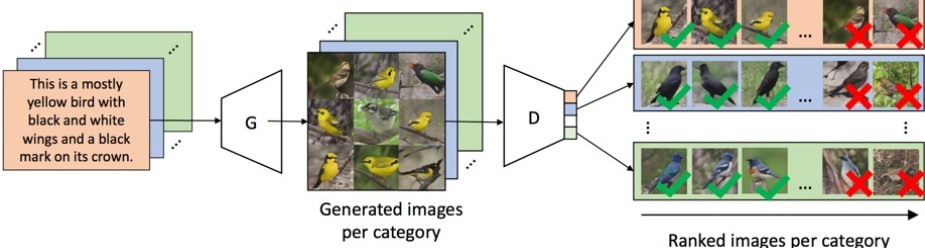

Figure 1: Selective image generation with tcGAN

**Discriminative Text-Conditional GAN:** Inspired by Wang et al. (2018), we follow a meta-learning framework and learn a generative model on the large amount of data available in $C_{Base}$, then utilize it to learn a classifier for limited sample in $C_{Novel}$. For that, we built a text-conditional GAN (tc-GAN) (e.g. Mirza & Osindero (2014), Reed et al. (2016b), Zhang et al. (2017)) to learn the mapping $\mathcal{T} \to \mathcal{I}$, which the generator $G$ is trained to produce outputs that cannot be distinguished from "real" images by an adversarially trained discriminator $D$, which is trained to do as well as possible at detecting the generators "fakes". This allows for cross-modal generation, which facilitates few-shot learning by compensating the lack of data in $C_{Novel}$.

Simplified, the objective of a tcGAN from observed text $T$ and image $I$ can be expressed as:

$$\mathcal{L}_{tcGAN}(G, D) = \mathbb{E}_{I,T}\left[\log D(I, T)\right] + \mathbb{E}_{I,z}\left[\log D(I, G(I, z))\right], \tag{1}$$

where $z$ denotes a random noise vector, and $T$ and $I$ observed text and image respectively.

In practice, we built our method on top of the StackGAN framework proposed by Zhang et al. (2017), a variant of tcGAN which features a robust pipeline for generating realistic images from fine-grained textual descriptions.

Optimization of $\mathcal{L}_{tcGAN}$ alone, however, lacks class-discriminativeness. Therefore we augment $\mathcal{L}_{tcGAN}$ by adding a class-discriminative term $\mathcal{L}_{disc}$, which is defined as:

$$\mathcal{L}_{disc}(D) = \mathbb{E}\left[P\left(C = c \mid I\right)\right], \quad \mathcal{L}_{disc}(D) = \mathcal{L}_{disc}(G), \tag{2}$$

where $c$ denotes the class label. This leads to two loss terms:

$$\mathcal{L}(D) = \mathcal{L}_{tcGAN}(G, D) + \mathcal{L}_{disc}(D), \quad \mathcal{L}(G) = \mathcal{L}_{tcGAN}(G) - \mathcal{L}_{disc}(G), \tag{3}$$

which are optimized in alternative fashion, yielding $D^*$ and $G^*$. It should be noted that whereas $\mathcal{L}_{tcGAN}$ is trained on samples from $C_{Base}$, the compound loss is trained only on the (sub-)set of $n$ training samples that are available within $C_{Novel}$. This provides us with the training of the GAN in a meta-learning fashion, where the cross-modal representation learned on the base classes, is later employed for the final class-discriminative few-shot learning task.

**Self-paced Sample Selection:** Training the text-conditioned GAN allows for the generation of a potentially infinite number of samples given textual descriptions using $G^*$. However, the challenge is to pick adequate samples out of the pool of generated samples that allow for building a

better classifier within the few-shot scenario. Such subset of images should not only be realistic but also class discriminative. To this end, we follow the *self-paced strategy* and select a subset of images corresponding to ones in which the generator is most confident about their "reality" and the discriminator is the most confident about their "class discriminativeness". Specifically, we use the score computed using $D^*$ per category and sort generated images in a descending order using these scores. Then we select the first N top-most elements. Intuitively, we select a subset of the generated samples that the classifier trained on the real data is most confident about, as illustrated in figure 1. Finally, a convolutional neural network (CNN) is trained on the concatenated set of real images and those ones selected as the best generated class-discriminative images.

## 3 EXPERIMENTAL RESULTS

For our experiments we use the CUB bird dataset (Wah et al. (2011)), which contains 11,788 images of 200 different bird species. The data is split equally in training and test data, resulting in roughly 30 training and 30 test images per category. 10 short textual descriptions per image are provided by Reed et al. (2016a). Following Zhang et al. (2017) we use the text-encoder pre-trained by Reed et al. (2016a) and split the data such that $|C_{Base}| = 150$, $|C_{Novel}| = 50$. To perform few-shot learning $n = \{1, 2, 5, 10, 20\}$ images of $C_{Novel}$ are used for training, as proposed by Hariharan & Girshick (2017). For the sake of simplicity, a CNN with basic architecture is employed for classification, although any other classifier is applicable. It consists of two convolutional layers paired with max-pooling, followed by two linear layers that are connected with dropout completed with a softmax of $|C_{Novel}|$ units. For training SGD is used for 800 epochs with a learning rate of 0.01 and momentum of 0.5. The experiments are composed of: **1)** *Single modality baseline (R):* we train the classifier only on real data, i.e. $n$ images per category. **2)** *Few-shot StackGAN baseline (StG):* we generated images using StackGAN Zhang et al. (2017) by conditioning on one caption randomly chosen (out of 10) for the missing $30 - n$ images of $C_{Novel}$, followed by training of the classifier on the extended dataset. **3)** *StGD baseline:* To show the importance of our proposed discriminative tcGAN, we generated images for captions using the $G$ of StackGAN ranked by the score of $D$ (real vs. fake discriminative) instead of $D^*$, retaining $30 - n$. **4)** *Our proposed method (StGD\*):* Similar to the prevoius baseline with the difference of employing the class-discriminative $D^*$ for ranking generated images (StGD$^*$). The top-5 accuracy of the classifier for our different experiments is reported in table 1. We observe that the proposed approach outperforms the baseline in the particular challenging few-shot scenarios with $n = 1, 2, 5$ by 4.9 to 8.6 percentage points, respectively. Additionally to commonly reported top-5 accuracy, we evaluated the experiments with top-1 and top-3 accuracy, observing a similar performance results. Using the score of $D$ as measure to rank the generated images alone has shown not to be sufficient. However, enforcing class-discriminativeness within the discriminator leads to significantly higher accuracies. Our experiments confirm that multi-modality allows to close the information gap in few-shot scenarios, yielding more robust classifiers.

Table 1: Top-5 accuracy in percent of our classifier for different setups (R, StG, StGD, StGD$^*$)

| n | 1 | 2 | 5 | 10 | 20 |
|---|---|---|---|----|----|
| Single modality baseline (R) | 19.9 | 24.8 | 36.8 | 49.2 | **70.6** |
| StackGAN  Zhang et al. (2017) (StG) | 27.3 | 30.7 | 37.4 | 45.3 | 69.0 |
| Our method w/o discriminative tcGAN (StGD) | 25.7 | 30.1 | 36.8 | 50.5 | 68.6 |
| Our proposed method - full (StGD*) | **28.5** | **31.6** | **41.7** | **52.2** | 68.5 |

## 4 CONCLUSION

In this paper, we proposed to extend few-shot learning to deal with multimodal data and introduced a discriminative tcGAN for sample generation along wiht a self-paced strategy for sample selection. Experiments on our proposed benchmark demonstrate that learning generative models in a cross-modal fashion facilitates few-shot learning by compensating the lack of data in novel categories. For future work we plan to investigate the use of $D^*$ as the final classifier. Furthermore, we seek to incorporate class-discriminativeness property at the representation learning stage jointly with the ranking loss of the self-paced stage.

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
