# OpenReview forum: "Multi-Modal Few-Shot Learning: A Benchmark"
_ICLR.cc/2018/Workshop — Reject_

### Official Review · AnonReviewer3 · 2018-03-09
**interesting idea**

**Rating:** 6
**Confidence:** 5

**Review:**

Though the idea of this paper has been explored in some degree in previous works, I still think the idea of this paper is interesting and novel enough

---

### Official Review · AnonReviewer2 · 2018-03-09
**ICLR Workshop Official Review**

**Rating:** 5
**Confidence:** 5

**Review:**

Summary: This paper paper calls itself “multi-modal few-shot learning”, but I think is more clearly summarised as addressing the task of few+zero-shot learning. By this I mean it addresses the scenario where recognisers for new categories are to be induced, and the available data is twofold: (i) a small number of training examples (few-shot assumption), and (ii) some metadata about the category in the form of a text descriptions of images of that category (text metadata is typical zero-shot assumption). The general idea is to train a text-conditional GAN to generate synthetic images of the new category, and these are used to augment the few-shot images, and thus train a more robust recognizer, and authors propose a couple improvements (class-discriminator, simple curriculum) to improve this. The results show it is better than naive few-shot learning.

Novelty: Borderline. The idea of combining few+zero shot settings has been done in some prior work ([A], among others), so the problem setting is not quite novel. The approach of data augmentation by GAN has been done in vision [B,C,D].
Clarity: Personally, I find the  spin of “multi-modal few shot” misleading. Then you expect the final model to be one that works on multi-modal data, but the final model only works on vision data. So I think the few+zero-shot explanation would be clearer.
Significance: Hard to know the empirical significance as comparison is very light. Existing state of the art few-shot and state of the art zero-shot methods are not compared. But it looks like the proposed few+zero shot method is much worse than prior methods that only use one of these cues, e.g., [E].
Quality: Would be better with some more controls like text-only baseline to complement image-only baseline.

[A] https://arxiv.org/abs/1801.09086
[B] https://arxiv.org/abs/1611.01331
[C] https://arxiv.org/abs/1711.00648
[D] https://arxiv.org/abs/1701.07717
[E] https://arxiv.org/abs/1711.06025

Pros/Cons:
+ Overall the paper is a reasonable idea which works to some extent.
- But it’s not very surprising. The novelty is borderline, and the empirical results are not great.
- Also the spin of the presentation obfuscates the problem setting really addressed.

---

### Official Review · AnonReviewer1 · 2018-03-10
**Proposes a novel split for existing dataset and an incremental method for muli-modal few-shot learning**

**Rating:** 4
**Confidence:** 4

**Review:**

The paper proposes to split existing dataset (CUB with text descriptions), into a few shots multi-modal learning setting. It also proposes to use a cross model generator based on GANs (which is also heavily inspired from previous works) to aid in training by compensating for the few shot setting.

The methodological contribution of the paper is highly incremental. The benchmark is a novel split on top of existing datasets. And hence, overall the contributions of the paper are not convincing enough. One suggestion could be that comprehensive benchmarking of many different methods be done on the task;  cross modal learning has been studied in many different contexts computer vision (eg. with sketches and real images for objects and faces).

---

### Decision · Program_Chairs · 2018-03-20
**ICLR 2018 Workshop Acceptance Decision**

**Decision:**

Reject

**Comment:**

Based on the reviews, this paper has not been accepted for presentation at the ICLR workshop. However, the conversation and updates can continue to appear here on OpenReview.